# Soil Chemical and Microbiological Properties Are Changed by Long-Term Chemical Fertilizers That Limit Ecosystem Functioning

**DOI:** 10.3390/microorganisms8050694

**Published:** 2020-05-08

**Authors:** Yong-Chao Bai, Ying-Ying Chang, Muzammil Hussain, Bin Lu, Jun-Pei Zhang, Xiao-Bo Song, Xia-Shuo Lei, Dong Pei

**Affiliations:** 1State Key Laboratory of Tree Genetics and Breeding, Key Laboratory of Tree Breeding and Cultivation of the State Forestry and Grassland Administration, Research Institute of Forestry, Chinese Academy of Forestry, Beijing 100091, China; baiychao@163.com (Y.-C.B.); yingying_chang0123@163.com (Y.-Y.C.); zhangjunpei@caf.ac.cn (J.-P.Z.); xiaobo.song@caf.ac.cn (X.-B.S.); leixiashuo@126.com (X.-S.L.); 2State Key Laboratory of Mycology, Institute of Microbiology, Chinese Academy of Sciences, Beijing 100101, China; muzammil0991@gmail.com; 3Yunnan Academy of Forestry, Kunming 650000, China; kmlubin@163.com

**Keywords:** walnut, fertilization, sustainable agriculture, soil chemical properties, microbial communities

## Abstract

Although the effects of fertilization and microbiota on plant growth have been widely studied, our understanding of the chemical fertilizers to alter soil chemical and microbiological properties in woody plants is still limited. The aim of the present study is to investigate the impact of long-term application of chemical fertilizers on chemical and microbiological properties of root-associated soils of walnut trees. The results show that soil organic matter (OM), pH_kcl_, total nitrogen (TN), nitrate-nitrogen (NO_3_^−^), and total phosphorus (TP) contents were significantly higher in non-fertilized soil than after chemical fertilization. The long-term fertilization led to excessive ammonium-nitrogen (NH_4_^+^) and available phosphorus (AP) residues in the cultivated soil, among which NH_4_^+^ resulted in soil acidification and changes in bacterial community structure, while AP reduced fungal diversity. The naturally grown walnut trees led to an enrichment in beneficial bacteria such as *Burkholderia*, *Nitrospira*, *Pseudomonas*, and Candidatus_*Solibacter*, as well as fungi, including *Trichoderma*, *Lophiostoma*, *Phomopsis*, *Ilyonectria*, *Purpureocillium*, *Cylindrocladiella*, *Hyalorbilia*, *Chaetomium*, and *Trichoglossum*. The presence of these bacterial and fungal genera that have been associated with nutrient mobilization and plant growth was likely related to the higher soil OM, TN, NO_3_^−^, and TP contents in the non-fertilized plots. These findings highlight that reduced chemical fertilizers and organic cultivation with beneficial microbiota could be used to improve economic efficiency and benefit the environment in sustainable agriculture.

## 1. Introduction

The unreasonable use of chemical fertilizers has negatively impacted the environment, caused food security issues, and reduced our dependency on the positive services that soil biodiversity provides for plant performance [1,2,3]. In conventional agriculture, however, chemical fertilizers are frequently used to obtain higher crop yields, but only 10%–40% of the fertilizers applied can be directly absorbed and used by plants. The remaining fertilizers in the soil are in the form of insoluble inorganic salts or leached into adjacent rivers, which is considered a major threat to global soil biodiversity [4,5,6,7]. The soil microbiota plays important roles, such as participating in the biogeochemical cycling of soil nutrients [8,9,10], helping to withstand abiotic stresses [11,12], producing phytohormones that improve plant growth [13], and preventing infections by phytopathogens [14,15,16]. Similarly, the plants host microbes and release root exudates that serve as a food source for soil microbiota [17]. The use of plant growth-promoting rhizobacteria (PGPR) and fungi (PGPF) may provide a sustainable alternative to the use of chemical fertilizers [18,19]. 

Walnut (*Juglans* spp.) is an economically important tree for nut and wood products that is widely grown in many countries. Due to its oil content and quality, walnut is known as the “king of plant oils”. Unsaturated fatty acids account for more than 90% of the total fat content [20]. In recent years, the production and consumption of walnut have substantially increased owing to the high nutritional value based on its fat content. The growers sought to largely increase walnut yields with the consequent overuse of chemical fertilizers. This had consequences for the environment, thereby affecting soil quality and causing diseases like leaf spot, anthracnose, walnut blight, canker, and brown apical necrosis [4,21,22]. Altogether, this resulted in economic losses for *J. regia* L. trees by decreasing the amount and quality of harvested walnut nuts. To improve the planting efficiency of walnut and reduce its negative impact on the environment, organic cultivation of walnut trees with a reduced application of chemical fertilizer and enhanced microbiological function needs to be further developed. 

Previous studies have shown that plants that rarely have root hairs can promote the uptake of nutrients by establishing mutualism with soil microorganisms [23,24]. We hypothesize that the microbiota in natural ecosystems impacts nutrient uptake in trees and cycling of soil nutrients, but due to the lack of root hairs in walnut, a large number of chemical fertilizers in the cultivated orchard destroy the soil microbial community and soil quality, resulting in the weak nutrient absorption capacity of trees and lower soil fertility [19,25,26]. However, to our knowledge, few studies have reported on the ecological effects of long-term fertilization on microorganisms in the woody perennials, especially economic trees, despite the increased number of studies dealing with soil microbial community structure and function in several annual crop species (soybean [15]; barley [27]; rice [28]; maize [29]; wheat, oat and pea [30]). Moreover, the walnut genome has been sequenced [31,32,33], and we can thus now use genetic techniques to better study walnut–microbe interactions.

We investigated the relationship between soil chemical properties and microbial community of the rhizosphere and root endosphere of naturally grown walnut trees and those cultivated with chemical fertilization by high-throughput sequencing and structural equation model (SEM). Our study aims to determine (1) the soil chemical properties in natural and cultivated ecosystems, (2) the differences in the rhizosphere and root endosphere microbial community between natural and cultivated ecosystems, and (3) the key soil factors affecting microbial diversity and composition. The results of this study will provide a basis for rational fertilization management of perennial woody plants to promote sustainable agriculture and increase the benefits by reducing the planting cost of walnut.

## 2. Materials and Methods

### 2.1. Site Description

A long-term fertilization experiment that started in 2002 in southwestern China was used to evaluate the impact of chemical fertilizers on nuts yield, soil chemical properties, and microbial community composition. The region has a continental monsoon climate with an annual average temperature of 12–17 °C, and annual average precipitation of 856–1144 mm. In the same experimental area, uniformly growing trees were selected for naturally grown and cultivated trees in 2002. In the first year of fertilization (2002), the same amount of organic fertilizer (organic fertilizer; 5000 kg/acre, broadcast fertilization) was applied to naturally grown and cultivated trees. In addition, the cultivated trees received chemical fertilizers (compound-fertilizer, N:P_2_O_5_:K_2_O = 15:15:15; 80 kg/acre, furrow application) in autumn. From then on (2003–2017), the naturally grown trees did not receive any type of fertilizer, whereas the cultivated trees typically received chemical fertilizers in autumn each year. The area of naturally grown trees is approximately 3–6 km away from the cultivated tree area. Other tree management measures were conducted according to “the China walnut complex standardization system” (LY/T 3004-2018; http://www.forestry.gov.cn). Our early experiments (from 2002 to 2017) were carried out to evaluate the effects of long-term fertilization on the yield of walnut nuts. The analyses of the walnut yield for 3 consecutive years revealed that fertilization treatments had comparably similar nut yield (not significantly different, *p* > 0.05) that represented 232.13 ± 14.32 kg/acre for naturally grown and 250.07 ± 7.22 kg/acre for cultivated trees.

### 2.2. Soil Sampling and Walnut Roots Collection

The collection of soil and root samples from naturally grown and cultivated walnut trees was conducted at five locations that are the main concentrated distributed area of walnut (Chuxiong, Dali, Honghe, Lincang, and Zhaotong) in southwestern China (Figure 1). Fifteen healthy walnut trees (five trees each) at the reproductive stage were randomly selected from three replicated locations of naturally grown and cultivated trees sites in each of five geographical locations in 2018. The soil samples for each site were collected from 20 to 30 cm soil depth, mixed thoroughly, and passed through a 2-mm mesh to remove the plant debris and stones (Appendix A). The samples were then stored in a cool box and transported to the laboratory for the analysis of soil chemical properties. The root samples were collected (20–30 cm depth; 1–2 mm diameter) by first clearing surface debris from the base of the walnut tree and then excavating the root with an ethanol-sterilized spade. For each tree, the fine roots (1–2 mm) were separated using sterile tweezers, and then placed into plastic bags and sealed for transport to the laboratory within a cooling box. The walnut root, which is fleshy and without root hairs, was observed by stereomicroscope (Appendix A). For rhizosphere soil, fine roots were shaken gently to remove excess soil, then roots were placed in 50 mL sterilized Falcon tubes containing 25 mL sterile Silwet L-77 amended PBS buffer (PBS-S: 130 mM NaCl, 7 mM Na_2_HPO_4_, 3 mM NaH_2_PO_4_, pH 7.0, 0.02% Silwet L-77), were washed on a shaking platform for 20 min at 180 rpm, and the rhizosphere soil was allowed to settle for 5 min. The roots were carefully removed and transferred to new 50 mL sterilized Falcon tubes, and the washing buffer was centrifuged at 10,000× *g* for 20 min. The supernatant was discarded carefully, and the rhizosphere soil was collected, frozen in liquid nitrogen, and stored at −80 °C until DNA extraction. For the root endosphere compartment, the epiphytes were first removed from the root surface, according to the method described by Beckers et al. (2017) [34] with some modifications. The roots were first washed with 70% (*v*/*v*) ethanol for 40 s, followed by 5% sodium hypochlorite (with active chloride ions) for 5 min, and then with 70% (*v*/*v*) ethanol for 30 s. Washed root samples were rinsed three times with sterile water and dried using sterile filter paper.

### 2.3. Soil Chemical Properties Analysis

Soil samples were air-dried and analyzed for soil pH (pH_KCl_) using a Thermo Orion-868 pH meter (Thermo Orion Co.; Waltham, MA, USA) in 1 mol·L^−1^ KCl system. Nitrate–nitrogen (NO_3_^−^), ammonium–nitrogen (NH_4_^+^), and available phosphorus (AP) were extracted with 0.01 M calcium chloride and quantified using an AutoAnalyzer 3 (SEAL Analytical GmbH, Norderstedt, Germany). The organic matter (OM) content was analyzed using the dichromate chemical oxygen demand test. All the soil samples were quantified as previously reported (Bao SD, Soil and Agricultural Chemistry Analysis, Beijing, Agriculture Publication (2000) p. 355-6).

### 2.4. Walnut Rhizosphere and Root DNA Extraction, PCR Amplification and High-Throughput Amplicon Sequencing

The MoBio PowerSoil DNA isolation kit (MoBio Laboratories, Carlsbad, CA, USA) was used for the extraction of DNA by following the manufacturer’s instructions. A total of 20 rhizosphere (2 replicates from each naturally grown and cultivated tree site were obtained due to very little soil being adhered to the fine roots of walnut) and 30 root endosphere samples were applied to extract genomic DNA by homogenizing the samples in the DNeasy instrument for 30 s at a speed setting of 6.0. The extracted DNA was eluted in 50 μL of elution buffer and then stored at −80 °C for further PCR amplification. The DNA concentrations were measured with a NanoVue Plus (BIOCHROM LTD, Cambridge, UK).

The DNA from each replicate was adjusted to 30 ng/μL for the rhizosphere and root endosphere from five locations. Fungal diversity in the rhizosphere and root endosphere was determined by sequencing the Internal Transcribed Spacer (ITS) region 2 of the fungal ITS gene with the specific barcoded primer pair ITS3F (5′-GCATCGATGAAGAACGCAGC-3′) and ITS4R (5′-TCCTCCGCTTATTGATATGC-3′) [35]. The bacterial community was characterized from the rhizosphere and root endosphere by sequencing the 16S rRNA V4 region with the specific barcoded primer pair 515F (5′-GTGCCAGCMGCCGCGGTAA-3′) and 806R (5′-GGACTACHVGGGTWTCTAAT-3′) [15]. The PCR amplifications were performed in triplicate on a Veriti thermal cycler (Applied Biosystems Veriti Thermal Cycler, Thermo Fisher Scientific, San Jose, MA, USA). The PCR products were then separated by agarose gel electrophoresis to purify the amplicons using a DNA Gel Extraction Kit (Takara, Kyoto, Japan) and pooled in equimolar concentrations before sequencing. Finally, paired-end sequencing of the bacterial and fungal amplicons was performed on an Illumina HiSeq sequencer at Guangdong Magigene Biotechnology Co. Ltd. (Guangzhou, China). All of the sequence data have been submitted to the National Genomics Data Center under BioProject ID PRJCA002614.

### 2.5. Bioinformatic and Statistical Analyses

Bacterial and fungal sequences were quality-trimmed using Trimmomatic v0.36 and assigned to samples based on barcodes using Quantitative Insights into Microbial Ecology (QIIME). De novo and reference-based chimera were checked, and sequences characterized as chimeric were removed. Bacterial and fungal reads were binned into operational taxonomic units (OTUs) at the ≥97% sequence similarity level using an open-reference OTU picking protocol in the UPARSE-pipeline [36], and the most abundant sequences from each OTU were taken as representative sequences of that OTU. Finally, the taxonomic configuration of bacterial and fungal OTUs was performed using the SILVA and UNITE databases, respectively. Bacterial OTUs representing the chloroplast or mitochondria were discarded before downstream analysis. The resulting OTU table was used to determine the taxonomic composition and for calculations of alpha- and beta-diversity. Alpha- and beta-diversity were calculated using QIIME (V1.9.1) and visualized using R software (ver. 2.1.5.3; R Development Core Team). Principal coordinate analysis (PCoA) was performed to analyze the similarities and differences among microbial communities, and linear discriminant analysis (LDA) and effect size (LEfSe) analysis were performed to identify the indicator taxa representing each group. For the LEfSe analysis, values were significant (*p* < 0.05) when the LDA score was more than 2. Diversity metrics and soil chemical properties were compared among samples with a one-way analysis of variance (ANOVA) using SAS software (ver. 9.1; SAS Institute, Cary, NC, USA). The least significant difference test was used to evaluate results and differences were considered significant at *p* < 0.05.

To identify the soil factors that have the main effects on microbiota and to provide a basis for the better management of agricultural intensification, a structural equation model (SEM) was employed to gain a mechanistic understanding of how soil chemical properties mediate alterations in soil bacterial community and fungal diversity [37]. The bacterial community (beta diversity) was obtained by principal component analysis (PCA), and the first principal component (PC1) and Shannon index of fungi (alpha diversity) were used for SEM analysis. All variables were standardized by Z transformation (mean = 0, standard deviation = 1). In the SEM analysis, OM is one of the main energy sources of soil microorganisms, and naturally grown trees receive OM but did not receive chemical fertilizers, whereas cultivated trees received OM in the first year and then chemical fertilizer afterwards. It was therefore assumed that fertilizers alter soil chemical properties, which in turn affects microbial community and diversity. Thus, the theoretical model assumed that (1) OM directly influences bacterial community and fungal diversity, (2) OM indirectly affects the bacterial community and fungal diversity by changing soil chemical properties, and (3) the rhizosphere directly influences the endosphere. Amos software (ver. 22.0, IBM/International Business Machines Corporation, Armonk, NY, USA) was used to construct the SEM in this study.

## 3. Results

### 3.1. Soil Chemical Properties

Soil chemical properties were distinct between naturally grown walnut trees and cultivated trees planted on field soils (Figure 2; ANOVA and LSD *p* < 0.05). In general, OM, pH_kcl_, TN, NO_3_^−^, and TP contents were significantly higher in natural soil than cultivated soil. However, the AP and NH_4_^+^ were significantly lower in natural soil than cultivated soil, except for at LC and ZT. Furthermore, the long-term application of chemical fertilizers in cultivated trees led to a decrease in soil pH_kcl_ that resulted in soil acidification.

We further estimated soil chemical properties of fertilized and non-fertilized areas according to the standard classification of soil nutrient levels in China (Appendix A). OM was in Grade 2–3, TN (DL up to 4) and TP were in Grade 5–6, AP was in Grade 1 in the fertilized soil, whereas OM was in Grade 1 (ZT drops to 2), TN in Grade 3 (CX up to 2), TP in Grade 2–3 (ZT drops to 4) and AP in Grade 2 (CX, DL up to 1) in non-fertilized soil.

### 3.2. Differences in Bacterial and Fungal Diversity Associated with Naturally Grown and Cultivated Trees

The within-sample diversity (α-diversity) of the bacterial and fungal communities from the rhizosphere and root endosphere was estimated using Shannon’s diversity index (Figure 3A,B). The Shannon diversity index values for the rhizosphere and root endosphere bacterial community were not significantly different between naturally grown and cultivated trees (LSD, *p* > 0.05), indicating that chemical fertilization had little effect on the bacterial diversity. On the other hand, the Shannon diversity values for the rhizosphere and root endosphere fungal community were significantly higher for naturally grown trees than for cultivated trees (Figure 3B; LSD, *p* < 0.05).

PCoA based on unweighted UniFrac distances, a metric that measures the phylogenetic relatedness of the whole community and is well suited for comparing beta-diversity patterns between complex microbial communities, was used to investigate bacterial and fungal beta-diversity in naturally grown and cultivated tree walnut system. Bacterial beta-diversity differed between the rhizosphere and root endosphere, as well as between naturally grown and cultivated trees. Principal components 1 (PC1) and 2 (PC2) explained 44.1% and 19.1% of the total variation, respectively (Figure 4A), indicating that chemical fertilization had important effects on the bacterial community composition. Similarly, the fungal community also differed between the endosphere and the rhizosphere, with PC1 and PC2 explaining 50.5% and 21.4% of the total variation, respectively (Figure 4B). These results indicated that chemical fertilization had an important effect on the microbial community structure and diversity of the cultivated walnut trees.

### 3.3. Dominant Bacterial Taxa in Walnut Rhizosphere and Root Endosphere

We obtained 798,445 quality-filtered reads of 16S rRNA corresponding to 21,681 OTUs in samples from cultivated trees, and 735,546 quality-filtered reads corresponding to 21,442 OTUs in samples from naturally grown trees (Appendix A). These bacteria OTUs belonged to 22 phyla (Figure 5A); Proteobacteria, Acidobacteria, Planctomycetes, Actinobacteria, and Bacteroidetes were the dominant phyla in the rhizosphere and root endosphere of the walnut trees. Bacterial community composition was found to be distinct between the rhizosphere and root endosphere. The rhizosphere of naturally grown and cultivated trees were composed of Proteobacteria (54% in naturally grown trees; 44% in cultivated trees), Acidobacteria (10% in naturally grown trees; 12% in cultivated trees), Planctomycetes (6% in naturally grown trees; 6% in cultivated trees), and Actinobacteria (4% in naturally grown trees; 5% in cultivated trees). Similarly, the root endosphere of naturally grown and cultivated trees was composed of Proteobacteria (63% in naturally grown trees; 62% in cultivated trees), Acidobacteria (5% in naturally grown trees; 7% in cultivated trees), Planctomycetes (15% in naturally grown trees; 4% in cultivated trees), and Actinobacteria (7% in naturally grown trees; 10% in cultivated trees). The relative abundance of phylum Proteobacteria was increased in the root endosphere than the rhizosphere of naturally grown and cultivated walnut trees. In contrast, Acidobacteria were decreased in root endosphere than the rhizosphere of naturally and cultivated walnut trees.

We then analyzed the relative abundance of bacteria in the rhizosphere and root endosphere of naturally grown and cultivated trees at the genus level (Figure 6A). The 30 most abundant genera associated with both naturally grown and cultivated trees belonged to 9 bacterial phyla. *Burkholderia* was the most abundant genus in the rhizosphere and root endosphere of both tree types. *Pseudomonas* and Candidatus_*Solibacter* were abundant in the rhizosphere and root endosphere of naturally grown trees, whereas *Pirellula*, *Bdellovibrio*, *Klebsiella*, *Acidothermus*, and *Flavobacterium* were abundant in the rhizosphere and root endosphere of cultivated trees. Cluster analysis results revealed that the bacterial communities differed between naturally grown and cultivated trees, implying that the cultivation process affected the bacterial community composition.

The LEfSe analysis was used to identify the main taxa that contributed to community differences. We identified 55 indicator taxa belonging to 10 bacterial phyla (Figure 7A) that significantly distinguished the bacterial community associated with cultivated trees from the community associated with naturally grown trees (*p* < 0.05, LDA score > 2). Eight genera within *Bacteroidetes*, *Gemmatimonadetes*, *Nitrospirae*, and *Patescibacteria* were indicators of the bacterial rhizosphere community of cultivated trees, whereas *Gemmatimonas* and *Nitrospira* within Gemmatimonadetes were indicators of the bacterial rhizosphere community of naturally grown trees. Seven phyla and eight genera distinguished the bacterial rhizosphere community of naturally grown trees from that of cultivated trees. Specifically, Candidatus_*Solibacter* and *Nitrospira* were more abundant in the rhizosphere of naturally grown trees. Conversely, four phyla and three genera distinguished the bacterial rhizosphere community of cultivated trees from that of naturally grown trees. Of these, *Bdellovibrio* and *Pajaroellobacter* were more abundant in the rhizosphere of cultivated trees. Additionally, two taxa within Actinobacteria and Cyanobacteria, including the family Moraxellaceae, were indicators of the bacterial endosphere community of cultivated trees, whereas two orders within Proteobacteria, Pseudomonadales and Betaproteobacteriales were indicators of the bacterial endosphere community of naturally grown trees (Appendix A).

### 3.4. Dominant Fungal Taxa in the Walnut Rhizosphere and Root Endosphere

The four dominant fungal phyla in the rhizosphere and root endosphere were Ascomycota, Basidiomycota, Glomeromycota, and Zygomycota (Figure 5B). Ascomycota comprised 43% and 44%, and Basidiomycota comprised 25% and 14% of the fungal communities in the root endosphere of naturally grown and cultivated trees, respectively. Similarly, Ascomycota comprised 64% and 60%, and Basidiomycota comprised 9% and 3% of the fungal communities in the rhizosphere of naturally grown and cultivated trees, respectively. Specifically, Ascomycota was dominant in the rhizosphere, and Basidiomycota had a higher relative abundance in the root endosphere. Moreover, the relative abundance of Basidiomycota was higher in the rhizosphere and root endosphere of the naturally grown trees than the cultivated walnut trees.

The 30 most abundant fungal genera in all samples belonged to Ascomycota, Basidiomycota, and Zygomycota (Figure 6B). The rhizosphere of naturally grown trees was mainly dominated by *Trichoderma*, *Lophiostoma*, *Phomopsis*, *Ilyonectria*, *Purpureocillium*, *Cylindrocladiella*, *Hyalorbilia*, *Chaetomium*, and *Trichoglossum.* These genera were relatively less abundant in the rhizosphere of cultivated trees. *Lophiostoma* and *Purpureocillium* were the most abundant genera in the root endosphere of both tree types. In addition, the LEfSe analysis identified 36 indicator taxa for the fungal communities associated with walnut trees (Figure 7B). For instance, *Tremellomycetes*, *Agaricomycetes,* and *Mortierella* were less abundant, whereas *Trichocladium*, *Trichoderma*, and *Cylindrocladiella* were more abundant in the rhizosphere of naturally grown trees than in the rhizosphere of cultivated trees. These differentially abundant taxa could be regarded as potential microbe biomarkers (Appendix A).

### 3.5. Influential Factors on Bacterial Community and Fungal Diversity

To determine the interactive effect of soil chemical properties on bacterial community and fungal diversity, an SEM was constructed, and the relationship models for the OM, pH_KCl_, NO_3_^−^, NH_4_^+^, and AP relative to the bacterial community and fungal diversity are shown in Figure 8. The results showed that this model was non-recursive, and accounted for 38% of the variation in pH_KCl_, 26.2% in AP, 27.1% in NO_3_^−^, 43.7% in NH_4_^+^, and 28.9% and 71.5% in the bacterial community of the rhizosphere and root endosphere, respectively (Figure 8A). The improved model had a good fit to the data (χ^2^ = 0.796, chi-square; *p* = 0.496, GFI = 0.918, goodness-of-fit index; AIC = 52.33, Akaike information criteria; and RMSEA = 0.000 < 0.05, root-mean-square errors of approximation), which also indicates that the hypothesis model was well adapted to the observation data. This model shows that there was a negative relationship between the NH_4_^+^ and the bacterial community of the rhizosphere and root endosphere, with correlation coefficients of −0.621 and −0.825, respectively. This indicates that fertilization significantly affected the bacterial community of the rhizosphere and root endosphere because of the increased NH_4_^+^ content and decreased soil pH_KCl_. This model also showed that there was a negative relationship between the NH_4_^+^ and pH_KCl_, with a correlation coefficient of −0.615, and a positive relationship between the OM and pH_KCl_ with a correlation coefficient of 0.617. This indicates that natural ecosystems had a higher OM content than cultivated systems, which indirectly affected the bacterial community of the rhizosphere and root endosphere by pH_KCl_. This model further indicates that there was a positive interaction between the rhizosphere and root endosphere bacterial community, with a correlation coefficient of 0.515 (*p* < 0.05).

We further quantified the contribution of each potentially influential factor (including OM, pH_KCl_, NO_3_^−^, NH_4_^+^, and AP) to the significant reduction (*p* < 0.05) in the fungal diversity in the rhizosphere and root endosphere induced by fertilization, when compared with naturally walnut trees during non-fertilization. The improved model proved a good fit to the data (χ^2^ = 0.795, chi-square; *p* = 0.496, GFI = 0.938, goodness-of-fit index; AIC = 52.39, Akaike information criteria; and RMSEA = 0.000 < 0.05, root-mean-square errors of approximation), and the results showed that the model was reasonable and acceptable (Figure 8B). There was a positive relationship between the OM and the fungal diversity of the rhizosphere and root endosphere, with correlation coefficients of 0.641 and 0.691. This result indicates that the natural ecosystems significantly affected the fungal diversity between the rhizosphere and root endosphere because of the increased OM content and improved soil acidity. AP was the dominant control on fungal diversity in the rhizosphere and root endosphere, with correlation coefficients of −0.59 and −0.568. This indicates that the cultivated walnut trees typically receive phosphorus fertilizer, which is linked to a reduction in fungal diversity. This model further indicates that there was a positive interaction between the rhizosphere and root endosphere fungal community, with a correlation coefficient of 0.755 (*p* < 0.05).

## 4. Discussion

Soil nutrients are the foundation of plant growth, but the addition of large amounts of chemical fertilizers to the soil results in environmental pollution and higher planting cost [25]. Unlike other woody plants, economic trees require a lot of energy to produce large amounts of nuts in addition to maintaining normal tree growth. Therefore, good soil structure and soil fertility are essential for stable and high nut yields. However, our understanding of the chemical fertilizers to alter soil chemical and microbiological properties in woody plants is still limited. Our early experiments to evaluate the effects of long-term chemical fertilization on the yield of walnut nuts for 3 consecutive years indicated that the chemical fertilization treatments had comparably similar nuts yield for cultivated trees (250.07 ± 7.22 kg/acre; not significantly different, *p* > 0.05) to naturally grown walnut trees (232.13 ± 14.32 kg/acre).

The microbial community of annual crops varies greatly due to the short growth period [38,39], while that of perennial woody plants may be stable due to the longer life cycle [40]. The microbial community composition has shown strong responses to site location, soil depth, and may be inherited in certain plant species [41,42]. In the current study, our results revealed high bacterial and fungal diversity in both the rhizosphere and root endosphere of walnut trees (Figure 3); however, the bacterial diversity in the rhizosphere samples was apparently more (but not significantly different) than in the root endosphere (Figure 3A). The rhizosphere has been reported to contain 10^11^ microbial cells per gram of root biomass and more than 30,000 bacterial OTUs [43]; therefore, the diverse microbes inhabiting the rhizosphere have a key role in plant nutrition, growth promotion, and disease protection [44,45,46,47,48]. We found that long-term fertilization significantly influenced the bacterial community composition (Figure 4A) but did not influence its diversity (Figure 3A) relative to non-fertilization, and similar conclusions have demonstrated that inorganic nitrogen fertilization changes soil bacterial (and fungal) community composition [49]. Proteobacteria, Actinobacteria, and Acidobacteria were identified as the predominant bacterial phyla in all samples (Figure 5A), and these predominant bacterial phyla have been reported to improve nitrogen use efficiency and promote plant growth [39,50,51]. The proportion of Proteobacteria was higher in the rhizosphere and root endosphere of naturally grown trees than cultivated trees, and Actinobacteria and Acidobacteria were less diverse in the endosphere of naturally grown trees than in that of cultivated trees. This result shows that the roots of naturally grown walnut recruited a higher proportion of nitrogen cycle-related bacteria and PGPR (Figure 5A and Figure 6A), indicating that the nitrogen transformation process is probably more efficient in the root environment of naturally grown than cultivated trees. We speculate that beneficial bacteria transform organic nitrogen into nitrate and ammonium so that it can be efficiently absorbed by walnut roots in natural ecosystems. It has been previously concluded that the abundance of Proteobacteria significantly increases in response to the addition of chemical fertilizer [52], which differs from our results. High levels of N fertilization, especially NH_4_^+^, can drive soil acidification as a result of the ammonium ion hydrolyzes to increase soil hydrogen ion and, thus, the negative effects of N addition on the soil acidity [53]. It is well-known that soil acidification often causes reduced soil nutrient availability and destroys the soil structure [54,55]. As we discussed, the application of long-term chemical fertilizer significantly altered soil fertility and the abundance of beneficial microbiota, while avoiding the use of chemical fertilization increased the OM, TN, NO_3_^−^, and TP contents (Figure 2) in naturally grown trees. Therefore, managing the beneficial microbiota is crucial for litter decomposition and the provision of soil OM and mineral nutrition to plants [56]. In conclusion, this study has shown that long-term fertilization directly and indirectly affects soil fertility and microbial community composition, specifically that Proteobacteria are more diverse in the rhizosphere and root endosphere of the naturally grown trees and Acidobacteria in cultivated trees, which is consistent with the findings of Castro et al. [57] and Gottel et al. [58]. Therefore, chemical fertilizers will not increase the abundance of Proteobacteria unless it improves soil fertility and vice versa. These findings support the hypothesis that soil microbiota provide an important ecosystem service in plant nutrient uptake and soil nutrient cycle in reasonable agricultural ecosystems.

Furthermore, at the genus level, the bacterial community in the rhizosphere and root endosphere differed between naturally grown and cultivated walnut trees (Figure 6A). The LEfSe analysis revealed several taxa to be keystone taxa in naturally grown and cultivated walnut trees, which was linked to special functions. Some of the taxa that were more abundant when associated with naturally grown trees are important to walnut health and growth. For example, *Nitrospira* spp. are key nitrite-oxidizing bacteria [59], *Pseudomonas* spp. are rhizobacteria that promote plant growth [60], Candidatus_*Solibacter* bacteria decomposes OM in the soil, thus improving soil quality [61], and *Burkholderia* can be used as a biological pesticide to prevent infections by phytopathogens [62] in naturally grown trees. It has been reported that the relative abundance of *Burkholderia* and *Nitrospira* spp. were greater in non-fertilized soil than in fertilized soil [63,64]. In addition, Pseudomonadales, which have previously been reported to be keystone taxa in the endosphere, carry out symbiotic nitrogen fixation and provide nitrogen to plants [65]. In conclusion, the root microbiota become one of the major determinants of plant growth in the naturally grown trees [66,67]. In contrast, bacteria that negatively affect plant growth, such as *Moraxellaceae*, *Pirellula*, and *Bdellovibrio*, are abundant in the rhizosphere and root endosphere of cultivated trees [68]. These results reveal that natural ecosystems might have availed nitrogen fixation and enriched specific groups of nitrogen fixers, whereas cultivated trees might have selected against beneficial microbiota but enriched pathogens. Taken together with previous research, our results reveal that long-term avoidance of chemical fertilization was more favorable than fertilization in shaping the beneficial bacterial community composition and improving soil fertility for perennial economic trees, highlighting the vital importance of suitable management strategies. 

Fungal community composition in the citrus rhizosphere was reported to be similar among field samples [69]. In this study, however, we found that long-term fertilization significantly influenced fungal diversity (Figure 3B) but did not influence its community composition (Figure 4B) compared with non-fertilization. The Shannon diversity index was higher for fungal communities in the rhizosphere and root endosphere of naturally grown trees compared with those in the rhizosphere and root endosphere of cultivated trees. At the genus level, the relative abundance of fungi for *Trichoderma*, *Lophiostoma*, *Phomopsis*, *Ilyonectria*, *Purpureocillium*, *Cylindrocladiella*, *Hyalorbilia*, *Chaetomium*, and *Trichoglossum* (all Ascomycota) indicate they are sensitive to fertilization, as their proportion decreased in response to chemical fertilization. These fungi play an important role in tree growth, soil nutrient cycling, and disease protection. For instance, *Trichoderma* and *Purpureocillium* are potential biocontrol agents of plant pathogens [70,71], and some species of *Phomopsis* promotes the growth and yield of the host plant, reduces fertilization and can be used as a biocontrol agent [72,73]. The naturally grown walnut trees were not treated with chemical fertilizer, and thus the soil around these trees supported a higher fungal diversity, as previously observed by Bossio et al. [74] and Xu et al. [75]. Furthermore, the Shannon index of the endosphere was positively correlated with the Shannon index of the rhizosphere (*r*^2^ = 0.755 *). These results imply that the recruitment of microbes from the rhizosphere to the root endosphere is dependent on the microbes available in the rhizosphere [76,77]. Therefore, walnut status was correlated with the community composition of associated microbes, indicating that increases in walnut growth could be attributed to the higher nutrient uptake rates and stronger anti-pathogen defenses facilitated by the microbial community associated with naturally grown trees [78]. Thus, soil microbes play important roles in plant growth and nutrient availability. Although microbial functions can be taxon-specific [79], it remains necessary to determine whether it is the entire microbiome or specific interactions that affect plant health the most.

As mentioned above, long-term chemical fertilization can influence soil chemical properties by increasing NH_4_^+^ and AP contents, whereas non-fertilization increased the OM, TN, NO_3_^−^, and TP contents. These changes could be partly explained by soil microbiota in natural ecosystems, whose activities have shown to be critical to the function of these nutrient cycles and uptakes [8,80]. However, the fertilizer that is applied to the soil can be absorbed and transported by plants or remain in the soil [81], but excessive mineral fertilizers have been shown to change protist [82], bacterial [83], and fungal communities [84], and also affect soil enzyme activities [85]. Therefore, a lack of an increased plant yield after chemical fertilizer applications may not only be linked to the soil chemistry but also to the soil biology. Consequently, more research is needed to elucidate why fungal diversity was lower, and bacterial community composition was different in cultivated trees compared with natural trees. In our study, an SEM was used to evaluate relationships between microbiota and soil chemical properties (Figure 8). The NH_4_^+^ and AP contents had a significant negative relationship with the bacterial community and fungal diversity of both the rhizosphere and root endosphere (Figure 8), respectively. However, the OM content had a significant positive relationship with soil pH and the fungal diversity of both the rhizosphere and root endosphere. These results indicate that the natural ecosystems significantly affect fungal diversity between the rhizosphere and root endosphere through beneficial microbiota to increased OM content that reducing soil acidification [26]. Moreover, the long-term application of fertilizer in walnut orchards resulted in a large amount of NH_4_^+^ and AP residue, which changes the bacterial community and reduces fungal diversity. It is likely that some microbial taxa might have been negatively impacted by fertilizer input in cultivated areas, whereas more competitive taxa might have thrived in this environment, thus changing microbial community composition [86]. Thus, in the process of walnut planting, the application of chemical fertilizer should be moderated, which is consistent with the conclusion of Tahovská et al. (2020) [87], so that it does not cause soil acidification and the soil microbial community has time to adapt through structural and functional changes. Appropriate agricultural management of perennial woody plants should conserve soil microbiota and allow soil microbes to be recruited into the rhizosphere and root endosphere [88,89]. This information will help with developing better soil management practices for walnut management and elucidating the mechanisms of microbiota establishment in the walnut rhizosphere and root endosphere.

## 5. Conclusions

In our study, a comprehensive understanding of the impact of the long-term fertilization on the diversity and composition of rhizosphere and root endosphere microbiota has the potential to help guide and inform efforts to improve soil health in the face of increasing chemical fertilizer use. The results of this study reveal that there are different soil chemical and microbiological properties between natural and cultivated ecosystems. We found that long-term fertilization caused excessive NH_4_^+^ and AP residues and soil acidification, reduced fungal diversity, and altered the bacterial community structure. Our work not only provides evidence that microbiota composition is affected by fertilization but also shows that unfavorable and or pathogenic microbiota may increase. Moreover, the rhizosphere and root endosphere of naturally grown trees hosted a higher fungal diversity and more beneficial bacterial taxa, and its associated microbiota had specific roles in soil nutrient cycling and plant growth. We suggest that the development of an organic cultivation mode of reducing the application of chemical fertilizer and reasonable inputs of organic fertilizers according to the soil fertility level are likely to improve economic benefits while protecting the environment.

## Figures and Tables

**Figure 1 microorganisms-08-00694-f001:**
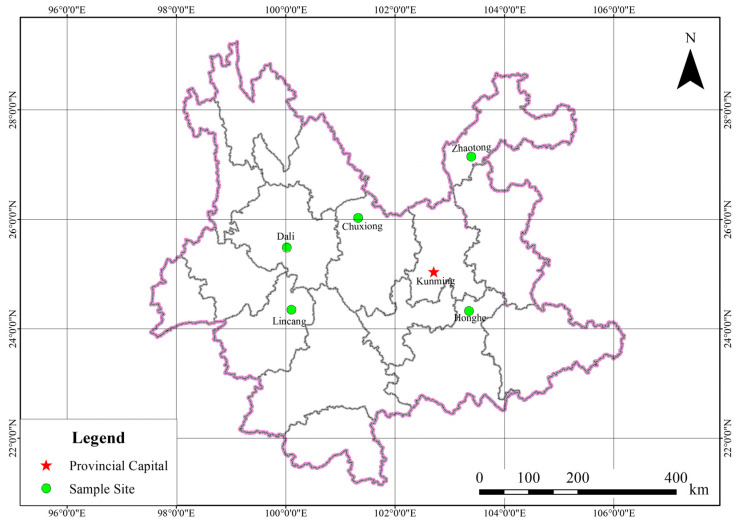
Map of sampling locations in the experiment areas in southwestern China. Diagram showing the collection of soil and root samples from naturally grown and cultivated walnut trees sites in each of five locations, which is indicated by the blue circle. The naturally grown and cultivated walnut trees areas are approximately 3–6 km apart, which were put in a circle in the same location. Fifteen healthy walnut trees (five trees each) at the reproductive stage were randomly selected from three replicated locations of a naturally grown and cultivated trees sites in each of five geographical locations.

**Figure 2 microorganisms-08-00694-f002:**
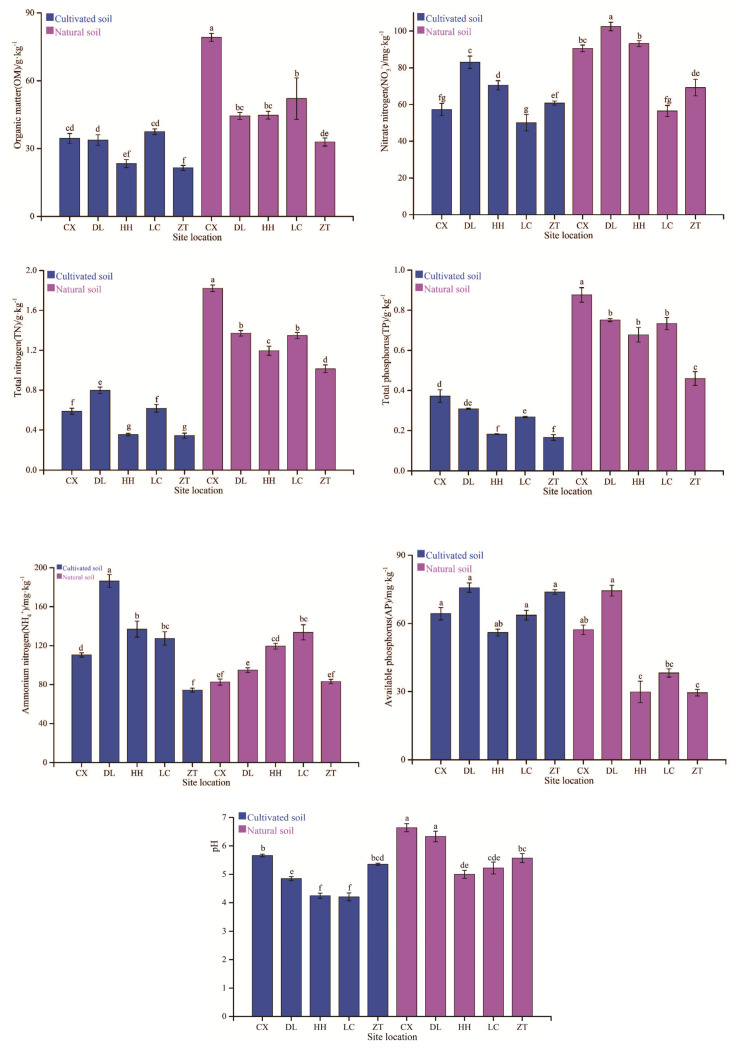
Soil chemical properties (mean ± SE, *n* = 3, error bars indicate the standard deviation) at naturally grown and cultivated walnut trees. Different lowercase letters on each column represent significant differences (*p* < 0.05) between the natural and cultivated soil based on the least significant difference test. The blue bar chart shows the soil chemical properties of cultivated soil after fertilization, and the brown bar chart shows the soil chemical properties of natural soil in non-fertilization. CX, Chuxiong; DL, Dali; HH, Honghe; LC, Lincang; ZT, Zhaotong.

**Figure 3 microorganisms-08-00694-f003:**
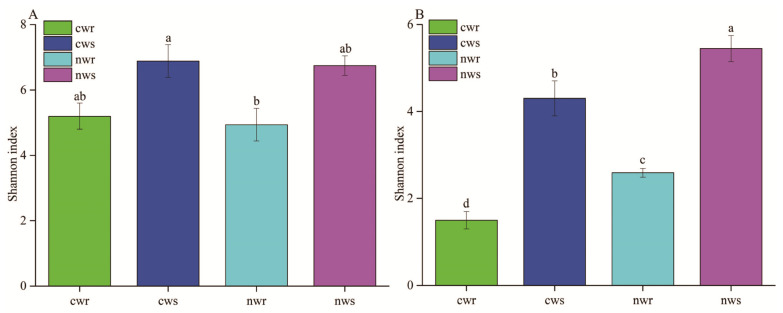
Shannon diversity indices of (**A**) bacterial and (**B**) fungal communities in the rhizosphere and root endosphere of naturally grown and cultivated trees. Different lowercase letters on each column represent significant differences (*p* < 0.05) between the rhizosphere and root endosphere based on the least significant difference test. cwr—cultivated tree endosphere; cws—cultivated tree rhizosphere; nwr—naturally grown tree endosphere; nws—naturally grown tree rhizosphere.

**Figure 4 microorganisms-08-00694-f004:**
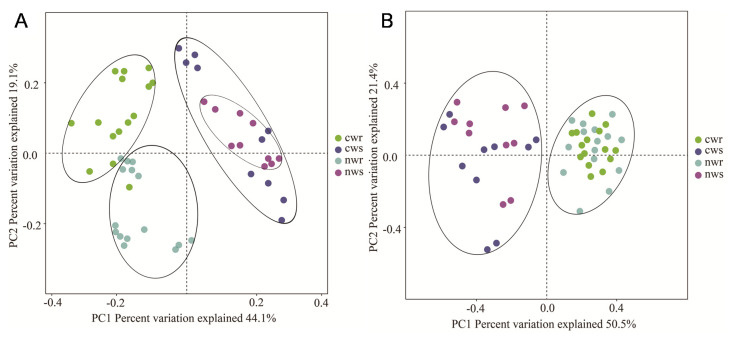
Principal coordinate analysis plot of the (**A**) bacterial and (**B**) fungal communities found in the rhizosphere and root endosphere of naturally grown and cultivated walnut trees. Principal components analysis (PCoA) plot of the first two principal components based on operational taxonomic units (OTUs) in the rhizosphere and root endosphere of walnut when exposed to naturally grown and cultivated walnut trees. 95% confidence ellipses (the ANOSIM test) were shown around the samples grouped based on naturally grown and cultivated trees. cwr—cultivated tree endosphere; cws—cultivated tree rhizosphere; nwr—naturally grown tree endosphere; nws—naturally grown tree rhizosphere.

**Figure 5 microorganisms-08-00694-f005:**
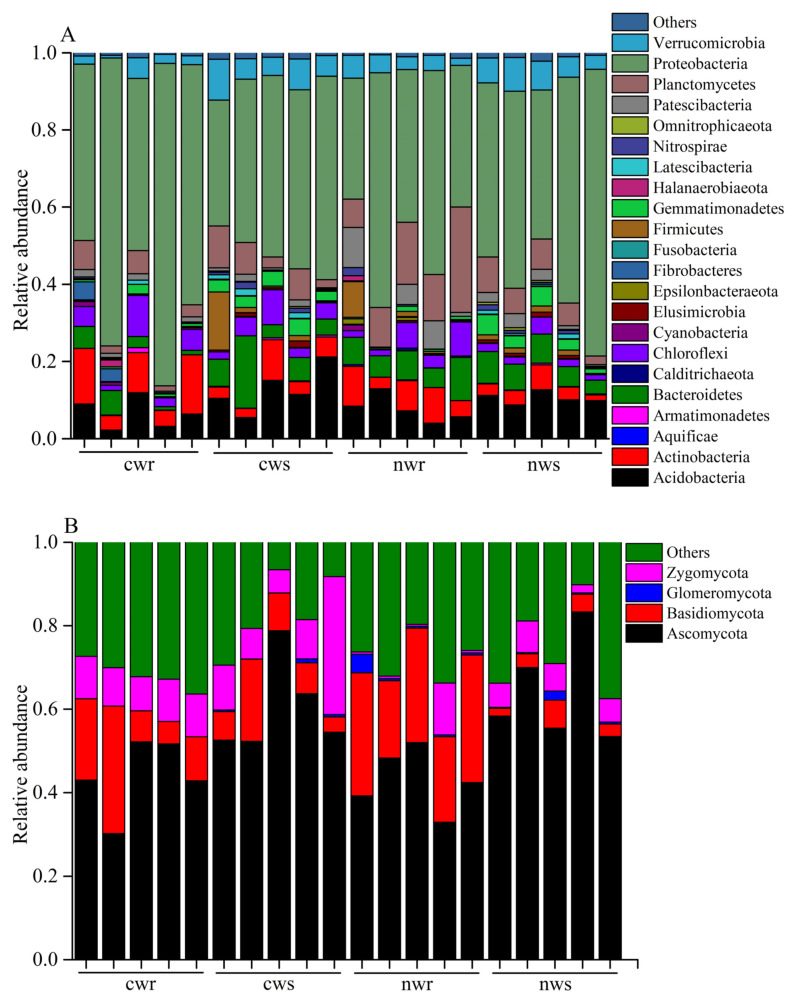
Relative abundance of (**A**) bacteria and (**B**) fungi at the phylum level in the rhizosphere and root endosphere of naturally grown and cultivated walnut trees. cwr—cultivated tree endosphere; cws—cultivated tree rhizosphere; nwr—naturally grown tree endosphere; nws—naturally grown tree rhizosphere.

**Figure 6 microorganisms-08-00694-f006:**
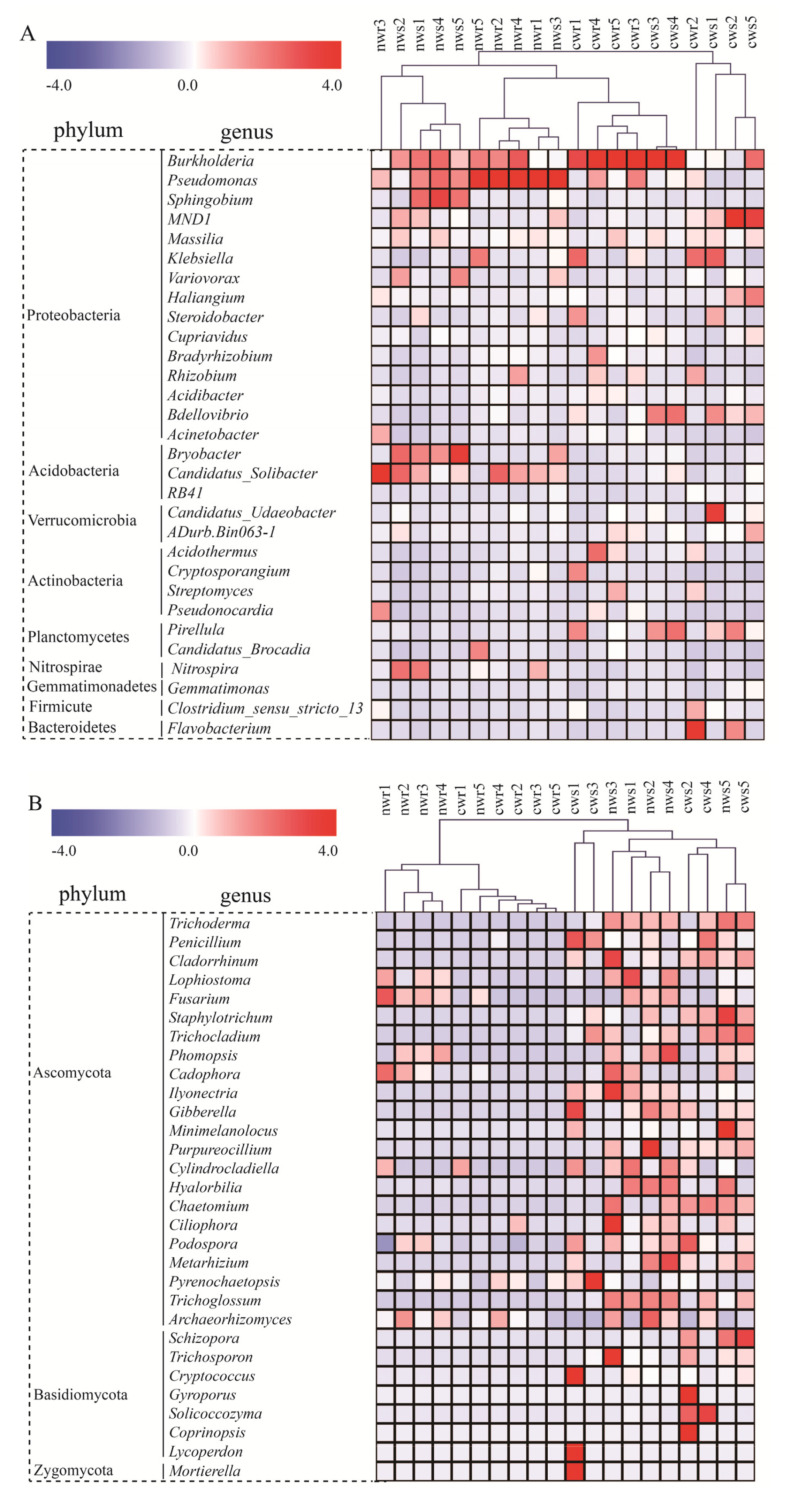
The 30 most abundant (**A**) bacterial and (**B**) fungal taxa in the rhizosphere and root endosphere across all sampling locations. Heatmap representing the hierarchical clustering of naturally grown and cultivated trees samples as well as relative changes in the abundance of bacterial and fungal genera inhabiting the rhizosphere and root endosphere of walnut. The red color indicates higher relative abundance, whereas the blue color indicates the lower the abundance.

**Figure 7 microorganisms-08-00694-f007:**
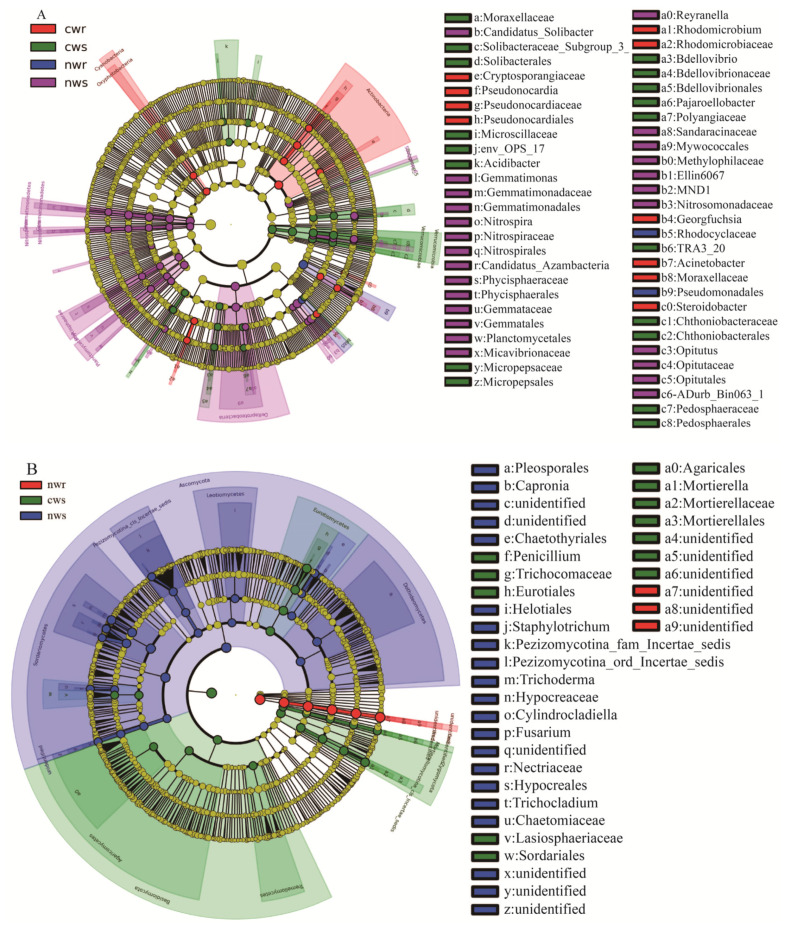
Bacterial (**A**) and fungal (**B**) taxa that contributed most to differences in microbial community composition. The linear discriminant analysis (LDA) effect size (LEfSe) analysis was performed to identify the indicator taxa representing each group, and the values were significant (*p* < 0.05) when the LDA score was more than 2.

**Figure 8 microorganisms-08-00694-f008:**
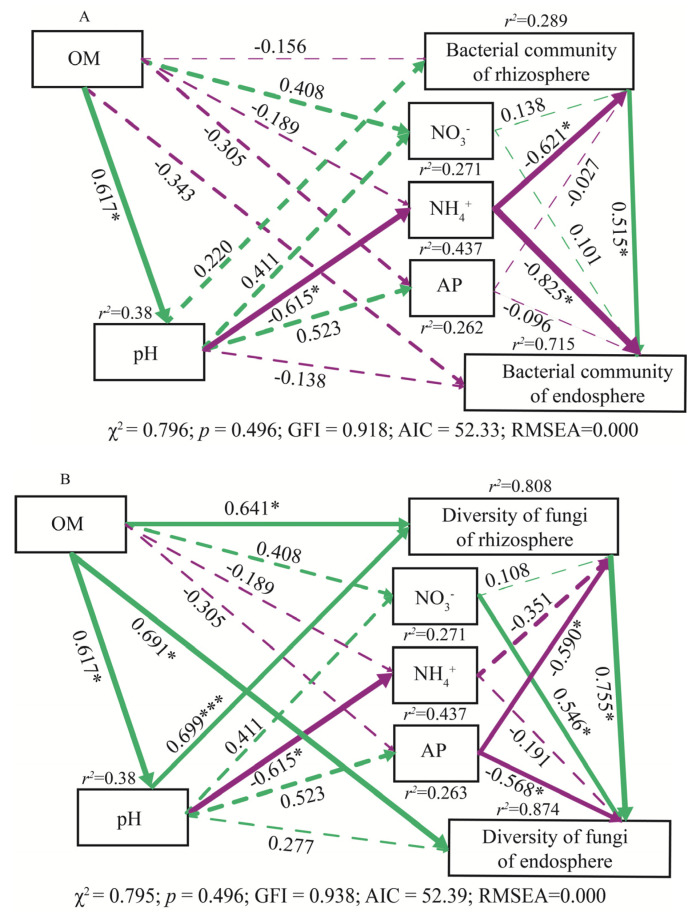
Structural equation model (SEM) shows influential factors of soil properties on the bacterial community (**A**) and fungal diversity (**B**). Green and brown arrows indicate positive and negative relationships, respectively. Significant levels are denoted * *p* < 0.05, ** *p* < 0.01, *** *p* < 0.001. Solid and dotted lines represent significant and insignificant differences, respectively. *r*^2^ values indicate the proportion of variance explained for each variable. The low chi-square (χ^2^), nonsignificant probability level (*p* > 0.05), high goodness-of-fit index (GFI > 0.90), low Akaike information criteria (AIC), and low root-mean-square errors of approximation (RMSEA < 0.05) listed below the SEMs indicate that our data matches the hypothetical models.

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
