# Peer review of "Soil Chemical and Microbiological Properties Are Changed by Long-Term Chemical Fertilizers That Limit Ecosystem Functioning"

_microorganisms, 2020, doi:10.3390/microorganisms8050694_

Round 1

Reviewer 1 Report

Please find my comments and suggestions in the attached document. 

Author Response

Dear reviewer, 

Thanks for your kind handling of the manuscript entitled “Soil chemical and microbiological properties were changed by long-term chemical fertilizers and limits ecosystem functioning”. We have thoroughly revised the manuscript based on your valuable comments and suggestions. A point-to-point response was also provided to facilitate you in locating the changes. Please see the attachment.

Thank you very much for your assistance

Sincerely yours,

Reviewer 2 Report

The research was done well in terms of methodology. The proportions between the individual sections are correct. Abstract, introduction, methodology, results are well presented.

The manuscript is printable after making the following corrections:

Line 103 has no spaces between 5000 and kg.

Line 105 and 108 did the proportions N: P: K relate to the phosphorus and potassium or elemental oxide form?

Figure 1 should be corrected. Village  names are not legible.

Lines 153, 154 and 545 minus and plus signs for ammonium and nitrate ions should be in superscript.

Figure 2 should be explain the abbreviations on the x-axis. Remove the bold dots on the y-axis.

Figure 3 should be explain the abbreviations on the x axis.

Figure 4 is too small font on the y and x axis.

Figure 5 The names of microorganisms are illegible. Explanations of abbreviations should be lowercase.

Figure 6 the font quality  is not good.

Figure 7 font quality should be improved.

Figure 8 not all numbers before the dot have zeros.

The References chapter should be adapted to the requirements of the magazine. For example, the titles of journals in items 6, 7, 14, 15, 19, 26, 32, 41, 42, 46, 49, 60, 66 and 78 have been abbreviated, and in the other items they are written with full names.

Author Response

Dear reviewer, 

Thanks for your kind handling of the manuscript entitled “Soil chemical and microbiological properties were changed by long-term chemical fertilizers and limits ecosystem functioning”. We have thoroughly revised the manuscript based on your valuable comments and suggestions. A point-to-point response was also provided to facilitate you in locating the changes. Please see the attachment.

Thank you very much for your assistance.

Sincerely yours,

Round 2

Reviewer 1 Report

The authors have carefully revised and addressed the comments posed in the first revision. Therefore, I find the present manuscript suitable for publication in the journal Microorganisms.

Yours faithfully,

María 

Author Response

Dear Professor María,

Thank you very much for the constructive comments and giving us an opportunity to revise our manuscript. Thanks a lot for your approval. 

Best Regards,

All the authors